# Influence of APOE4 Genotypes on Nutrient–Cognition Relationship in Taiwanese Older Adults: Longitudinal Findings from the HALST

**DOI:** 10.3390/nu18010106

**Published:** 2025-12-28

**Authors:** Rai-Hua Lai, Shiu-Ju Yang, Pei-Yi Hsu, Yi-Chung Chen, Shu-Chun Chuang, Chih-Cheng Hsu, Chao Agnes Hsiung, Fang-Lin Kuo

**Affiliations:** 1National Center for Geriatrics and Welfare Research, National Health Research Institutes, Yunlin 63247, Taiwan; lrh0330@nhri.edu.tw (R.-H.L.); shiujuyang@nhri.edu.tw (S.-J.Y.); 131206@nhri.edu.tw (P.-Y.H.); 120616@nhri.edu.tw (Y.-C.C.); 900613@nhri.edu.tw (C.-C.H.); 2Institute of Molecular and Genomic Medicine, National Health Research Institutes, Zhunan 35053, Taiwan; 3Institute of Population Health Sciences, National Health Research Institutes, Zhunan 35053, Taiwan; 011002@nhri.edu.tw (S.-C.C.); hsiung@nhri.edu.tw (C.A.H.)

**Keywords:** APOE genotype, cognitive function, dietary nutrients, aging population, Mini-Mental State Examination (MMSE), precision nutrition

## Abstract

**Background**: Older adults carrying the APOE4 allele are at elevated risk for cognitive decline. To clarify how dietary patterns may influence cognitive deterioration in this high-risk group, further investigation is needed. **Methods**: This prospective cohort study followed 1420 Taiwanese adults aged 65 years or older. Dietary intake was assessed using a validated food frequency questionnaire, and cognitive function was measured with the Mini-Mental State Examination (MMSE). Changes in 31 nutrients between two survey waves were used to simulate the effect of dietary shifts, and dietary patterns were derived using principal component analysis (PCA) with oblimin-derived scores. The analysis was further stratified by APOE genotype, and multiple linear regression models adjusted for demographic and health-related factors were applied to evaluate the associations between dietary changes and cognitive function. **Results**: Positive associations between dietary change and MMSE scores were observed only among APOE4 carriers. In this group, lower adherence to a plant-based pattern (TC1, estimate = 0.115, 95% CI = 0.029, 0.201) and higher adherence to an animal- and fat-rich pattern (TC2, estimate = −0.119, 95% CI = −0.202, −0.035) were both associated with poorer cognitive performance. **Conclusions**: APOE4 carriers may be particularly sensitive to dietary patterns, suggesting that genotype-informed nutritional strategies could help preserve cognitive health in older adults.

## 1. Introduction

Alzheimer’s disease (AD) is a multifactorial neurodegenerative disorder marked by considerable heterogeneity in its clinical course and underlying pathophysiological mechanisms [1,2]. Among the established genetic risk factors, the apolipoprotein E4 (APOE4) allele plays a pivotal role, contributing to amyloid-β accumulation and tau-related neurodegeneration [3,4]. However, for APOE4 carriers who are at elevated risk for cognitive impairment and dementia, there are currently no effective pharmacological treatments available to halt or delay disease progression. Therefore, lifestyle interventions are considered a potential alternative strategy for slowing cognitive decline [5].

Diet is one of the key modifiable determinants of late-life cognitive health. Several cohort studies demonstrate that the APOE4 genotype modulates the relationship between diet and cognition. For example, in the 12-year follow-up of the 3-City cohort, a higher dietary glycemic load was primarily associated with accelerated cognitive decline among APOE4 carriers [6]. In a Chinese longevity cohort, significant interactions between the diversity of dietary protein intake and APOE genotype were observed [7], while higher dietary fiber intake was especially protective for APOE4 carriers in a UK community study [8]. Beyond observational studies, several nutritional intervention trials have highlighted genotype-dependent effects. Medium-chain triglycerides (AC-1202), for example, primarily improved cognitive function in non-APOE4 individuals [9]. In high-dose docosahexaenoic acid (DHA) supplementation trials, APOE4 carriers exhibited a smaller increase in cerebrospinal fluid DHA, suggesting that higher or prolonged supplementation may be necessary to achieve effective brain exposure in this subgroup [10]. Evidence also suggests that Mediterranean-like dietary patterns and ω-3 fatty acids may confer greater protective benefits among APOE4 carriers [11]. However, substantial heterogeneity remains across studies, with wide variation in sample size, a predominant focus on single nutrients, and inconsistent responses to nutrient-specific interventions. These findings suggest that the influence of the APOE4 genotype on diet–cognition relationships is not uniform, but may be shaped by dietary patterns, cultural context, and population-specific metabolic characteristics.

Recent evidence from Taiwan shows that metabolic risk is highly prevalent among older adults [12], with over 40% meeting the criteria for metabolic syndrome. The most common risk profile included elevated blood glucose, hypertension, and central obesity combinations of these components were strongly linked to new-onset diabetes. Given this context, incorporating chronic diseases as a key factor is important when examining determinants of cognitive change in older adults.

Taiwan has a distinctive dietary culture and a diverse population background [13,14]. Among older adults, daily diets are characterized by staple carbohydrates such as rice and congee. These diets also feature high consumption of soy-based products, island-specific fruits, and various fermented foods. Together, these components form a dietary pattern distinct from that of Western populations. To better capture region-specific nutritional characteristics, we applied principal component analysis (PCA) with oblimin method to derive nutrient-based dietary patterns, representing habitual eating behaviors and age-related shifts in nutrient intake. In this longitudinal study of older adults in Taiwan, we examined how changes in nutrient intake over time relate to long-term cognitive trajectories and the risk of cognitive decline. Cognitive function was assessed using the Mini-Mental State Examination (MMSE). Furthermore, we conducted analyses stratified by APOE2, APOE3, and APOE4 genotypes. Nutrient changes between the two survey waves were modeled as the primary exposure. This approach allowed us to elucidate genotype-specific associations between dietary changes and cognitive outcomes in this culturally distinct aging population. We hypothesized that PCA-derived nutrient-based dietary patterns are associated with longitudinal cognitive change in Taiwanese older adults. We expected these associations to vary by APOE genotype.

## 2. Materials and Methods

### 2.1. Study Population

Participants were drawn from the Taiwan Longitudinal Study on Aging and Health (HALST), a community-based prospective cohort initiated in 2009 that enrolled adults aged 55 years and older. This study was a longitudinal prospective cohort analysis using data from the HALST study. The HALST cohort is registered at ClinicalTrials.gov (ID: NCT02677831). For the present analysis, we included only participants aged 65 years or older with complete data on dietary intake (food frequency questionnaire, FFQ), cognitive testing (MMSE scores between 18 and 30), and genotyping information. Wave 1 was conducted between 2009 and 2013, and Wave 2 between 2013 and 2017. Participants were recruited from seven geographic regions in Taiwan through multistage sampling. Participants were followed prospectively from Wave 1 to Wave 2 (*n* = 1420). Written informed consent was obtained from all participants. The study protocol was approved by the Institutional Review Board of the National Health Research Institutes, Taiwan (EC0970608, EC1020805, and EC1140803). The median follow-up interval was 6.38 years (IQR: 5.45–6.77), with a mean of 6.22 years (SD: 0.85; range: 4.49–9.52).

### 2.2. Dietary Assessment

Dietary intake was assessed using a validated food frequency questionnaire (FFQ) administered by trained interviewers. Participants reported their usual frequency and portion sizes of food and beverage consumption over the past year [15,16]. Nutrient intake was estimated using a comprehensive food composition database maintained by the Taiwan Food and Drug Administration, applying the 2020 nutrient conversion standards. Daily intake levels were calculated for 31 nutrients, including vitamins, minerals, fatty acids, and amino acids, following the approach described by SC Chuang et al. [13]. Participants with implausible total energy intake (<500 or >5000 kcal/day) were excluded from the analysis.

### 2.3. Cognitive Measurement

Cognitive performance was assessed using the Mini-Mental State Examination (MMSE), a widely used 30-point scale evaluating orientation, registration, attention and calculation, recall, and language [17] The MMSE was administered by trained research staff at both Wave 1 and Wave 2. Change in cognitive function (ΔMMSE) was defined as the difference in MMSE scores between the two waves/followed up years. To focus on individuals in the early stages of cognitive decline, such as those with possible mild cognitive impairment (MMSE 18–23) or normal cognition (MMSE 24–30) [17], participants with baseline MMSE scores < 18 were excluded.

### 2.4. APOE Genotyping

APOE genotype analysis was performed by genome-wide SNP array data generated from the TWBv2.0 platform (Axiom Genome-Wide Array Plate system, Thermo Fisher Scientific, Waltham, MA, USA) of HALST cohort participants. Genomic DNA was extracted from whole blood and analyzed using the TWBv2.0 SNP array. APOE genotype was determined based on the allelic variants at rs429358 and rs7412. A total of 1420 participants were classified into three genotype groups: APOE4 carriers (those with at least one E4 allele and no E2 allele), APOE3 homozygotes (E3/E3), and APOE2 carriers (those with at least one E2 allele and no E4 allele). Participants with the E2/E4 genotype (*n* = 19) were excluded from the analysis due to the presence of both risk and protective alleles, which precluded clear classification.

### 2.5. Principal Component Analysis of Nutrient Intake

To reduce dimensionality and derive interpretable nutrient groupings, we performed principal component analysis (PCA) on 31 nutrient intake variables measured at each study wave. All nutrient variables were standardized (mean = 0, SD = 1) to ensure comparability across different measurement scales. PCA was conducted on the correlation matrix, yielding principal components (PCs). After examining component intercorrelations, we applied an oblimin (oblique) rotation to obtain transformed components (TCs) that allowed for correlated dietary patterns. Components were retained to explain approximately 78.4% of the cumulative variance. Individual component scores (TC1–TC3) were computed and subsequently used as predictors in regression models to represent broader nutrient intake pattern profiles. Eigenvalues as the amount of variance explained by each principal component, and loading values as the strength and direction of the association between each original variable and a given component. All statistical analyses, including PCA, were performed in R (version 4.2.3, R Project for Statistical Computing) using the principal() function from the psych package.

### 2.6. Blood Collection and Biochemical Measurements

Participants fasted for 8–12 h prior to venipuncture. Blood samples were collected into serum separator tubes for lipid measurements and EDTA-containing tubes for vitamin measurements. Samples were processed on the day of collection according to the clinical laboratory’s standard operating procedures, including centrifugation to obtain serum for lipid assays and EDTA plasma for vitamin assays, and were subsequently transported to a certified clinical laboratory for analysis.

Serum lipid parameters, including total cholesterol, triglycerides, high-density lipoprotein (HDL) cholesterol, and low-density lipoprotein (LDL) cholesterol, were measured using automated clinical chemistry analyzers. During Wave I and Wave II, lipid measurements were performed using the Bayer ADVIA 1800 system, and following a 2017 equipment upgrade in Wave II, measurements were conducted using the Siemens ADVIA XPT system (Siemens Healthineers, Erlangen, Germany). Total cholesterol was quantified using a cholesterol oxidase method, triglycerides were measured by an enzymatic glycerol phosphate oxidase method based on the Trinder reaction without a serum blank, and HDL and LDL cholesterol were determined using homogeneous elimination and catalase-based assays.

Plasma concentrations of vitamin B12 and folate were measured using automated immunoassay platforms. During Wave I and Wave II, analyses were performed using the Bayer ADVIA Centaur system (Bayer Diagnostics, Tarrytown, NY, USA), and after the 2017 upgrade, the Siemens Centaur XPT system (Siemens Healthineers, Erlangen, Germany) was used. Both vitamin B12 and folate concentrations were quantified by chemiluminescence immunoassays. All laboratory measurements were performed in accordance with standardized operating procedures, with routine instrument calibration and internal quality control to ensure analytical accuracy and reliability.

### 2.7. Longitudinal Regression Analysis

To account for differences in the time interval between the two assessments, the change in all continuous variables, except for age, sex, education, was standardized by the number of years between waves. This approach allows for more accurate comparisons across participants with different follow-up durations. Based on PCA-derived component scores and APOE genotype, multiple linear regression models were used to predict the effects of these nutrient changes on ΔMMSE. The model adjusted for baseline age, sex, and education level, along with hypertension, diabetes, hyperlipidemia, total energy intake change. These variables were considered potential confounders based on prior literature. To examine the interaction between dietary patterns and APOE genotypes on cognitive performance, interaction terms between APOE genotype and each transformed component score (TC1–TC3) were included in separate regression models. Predicted changes in MMSE (ΔMMSE) were estimated from these models and visualized using the ggplot2 and ggeffects packages in R (version 4.2.3). Information on the measurement methods and characteristics of questionnaire-based and biomarker variables is provided in the Appendix A (Excel).

To account for multiple comparisons in genotype × nutrition component (TC) interaction tests, we applied the Benjamini–Hochberg false discovery rate (FDR) procedure. FDR-adjusted q values were calculated following the approach [18]. The study sample size was determined by the availability of eligible participants in the HALST cohort who had complete dietary, cognitive, and genetic data. A post hoc power analysis using G*Power (version 3.1.9.7; Heinrich Heine University Düsseldorf, Düsseldorf, Germany) indicated that the final sample size provided adequate statistical power to detect effects of the magnitude examined.

The Kaiser–Meyer–Olkin (KMO) measure of sampling adequacy was 0.87, and Bartlett’s test of sphericity was statistically significant (*p* < 0.001), indicating that the nutrients data were suitable for PCA. To minimize potential sources of bias, standardized protocols were used for dietary assessment, cognitive testing, and genotyping across study waves. Participants with implausible energy intake and ambiguous APOE genotypes were excluded to reduce measurement and classification bias. Only participants with complete data on dietary intake, MMSE, and genotyping were included in the analysis; no imputation for missing data was performed.

## 3. Results

### 3.1. Participant Characteristics

The study included 1420 participants in the longitudinal analysis (Figure 1). The genotype distribution in this subsample conformed to the Hardy–Weinberg equilibrium (rs429358: exact HWE, *p* = 0.874; rs7412: exact HWE, *p* = 0.335). Furthermore, the APOE genotype distribution observed in this study was comparable to that reported in the Taiwan Biobank, with no significant differences detected (rs429358: X^2^ = 0.331, *p* = 0.848; rs7412: X^2^ = 2.183, *p* = 0.336). These results indicate that the genetic composition of our study sample is representative of the general Taiwanese population.

Table 1 summarizes the baseline characteristics of participants stratified by APOE genotype. At baseline (Wave 1), the mean age was comparable across APOE4, APOE3, and APOE2 carriers (70.31, 70.45, and 70.73 years, respectively). The proportions of female participants were also similar among the three groups (57.4%, 54.6%, and 55.5%, respectively). Educational attainment did not differ meaningfully across genotypes, with average years of schooling of 8.34 ± 4.39 in APOE4 carriers, 8.21 ± 4.53 in APOE3 carriers, and 7.90 ± 4.51 in APOE2 carriers. Baseline MMSE scores were high and comparable across the three groups, ranging from 26.4 to 26.7 (Table 1).

During follow-up, modest declines in MMSE were observed in all genotypes (−0.31 in APOE4, −0.29 in APOE3, and −0.27 in APOE2 carriers) (Table 1). Reductions in macronutrient intake were also similar across groups, including decreases in total carbohydrates (−6.00, −2.80, and −2.56 g), total dietary fiber (−0.62, −0.47, and −0.44 g), total protein (−1.08, −0.67, and−0.64 g), and small increases in total fat intake (0.41, 0.33, and 0.17 g). Total energy intake likewise decreased slightly across groups (−24.24, −10.48, and −11.39 kcal, respectively) (Table 1). None of these differences reached statistical significance.

### 3.2. Principal Component Scores

Following the PCA, a total of 31 components (PC1 to PC31) were extracted (Appendix A). As shown in the scree plot (Appendix A), a clear inflection point emerged after the third component, indicating that PC1–PC3 captured the majority of meaningful variance (Appendix A). These three components had eigenvalues greater than 1 and collectively explained 78.4% of the total variance. PC1 accounted for 60.9% of the variance (eigenvalue = 18.88) and PC2 explained 12.9% of the variance (eigenvalue = 4.02). And PC3 accounted for 4.6% of the variance (eigenvalue = 1.42). After oblimin rotation, three transformed principal components (TC1–TC3) were interpreted as distinct dietary patterns (Figure 2). TC1 was characterized by higher intakes of folate, dietary fiber, vitamin C, and β-carotene, consistent with a plant-based, antioxidant-rich dietary pattern. TC2 loaded positively on fatty acids, cholesterol, and protein, reflecting an animal-based, fat-rich dietary pattern. TC3 was primarily driven by sugars and total carbohydrates, representing a carbohydrate-dominant dietary pattern. Overall, the PCA identified three clearly differentiated dietary patterns in this cohort. To evaluate whether the identified dietary patterns reflect participants’ actual nutritional status, we conducted additional analyses showing that TC1 aligned with circulating folate levels, and TC2 aligned with LDL cholesterol and total cholesterol (Appendix A). Overall, these findings support the validity of the PCA-derived dietary patterns as biologically meaningful indicators of nutrient intake and related metabolic profiles.

### 3.3. Main Effects of Demographic and Clinical Variables

In the multiple linear regression model predicting annualized change in MMSE score across two survey waves, age was significantly associated with cognitive decline (estimate = −0.018, *p* < 0.001) (Table 2), indicating that older individuals experienced greater decreases in MMSE over time. Education remained a strong protective factor (estimate = 0.029, *p* < 0.001) (Table 2). Among disease-related variables, baseline diabetes was significantly associated with greater cognitive decline (estimate = −0.094, *p* = 0.008) (Table 2). New-onset diabetes (first reported at Wave 2) was also negatively associated with cognition, with marginal statistical significance (estimate = −0.094, *p* = 0.065) (Table 2). In contrast, neither persistent nor new-onset hypertension or hyperlipidemia showed significant associations with cognitive change in this model. In the interaction analysis, significant effect modification by APOE genotype was observed for TC1 and TC2 (Table 3). Specifically, the interaction between APOE4 and TC1 was positively associated with cognitive outcome (estimate = 0.115, 95% CI: 0.029 to 0.201, *p* = 0.009, q = 0.026), whereas the APOE4 × TC2 interaction showed a significant negative association (estimate = −0.119, 95% CI: −0.202 to −0.035, *p* = 0.005, q = 0.026) (Table 3). In contrast, no significant interactions were observed for APOE2 with TC1 or TC2, nor for interactions involving TC3 across APOE genotypes (all *p* > 0.05 after multiple testing correction) (Table 3). These findings indicate that the associations of TC1 and TC2 with cognitive performance differed specifically among APOE4 carriers.

### 3.4. Principal Component Scores and Genotype Interactions

Principal component scores derived from nutrient variables (TC1, TC2, and TC3) alone were not independently associated with cognitive change (Table 2). Significant interaction effects were observed between APOE genotype and nutrient-based principal component scores. Specifically, individuals with the APOE4 genotype exhibited a significantly negative interaction with TC2 (estimate = −0.119, 95%CI = −0.202 to −0.035, *p* = 0.005, q = 0.026) and a positive interaction with TC1 (estimate = 0.115, 95%CI = 0.029 to 0.201, *p* = 0.009, q = 0.026), suggesting that nutrient patterns represented by these components may have differential impacts on cognition depending on genetic susceptibility (Figure 3).

## 4. Discussion

This study observed associations between changes in dietary nutrient intake and cognitive function among older adults in Taiwan during approximately six years of follow-up. Using principal component analysis (PCA), 31 nutrients were grouped into three distinct dietary patterns: TC1 (characterized by plant-based and antioxidant nutrients), TC2 (characterized by animal- and fat-rich foods), and TC3 (characterized by carbohydrate-rich nutrients) (Figure 2). The primary findings revealed that positive associations were observed only among APOE4 carriers: individuals with higher TC1 scores exhibited a slower rate of cognitive decline, whereas those with higher TC2 scores showed a trend toward accelerated decline. In contrast, no comparable associations were found among APOE3 or APOE2 carriers. These findings suggest that APOE4 carriers are particularly sensitive to dietary influences on cognitive trajectories, underscoring the potential of dietary modification as a precision strategy to delay cognitive decline in this genetically susceptible population.

Notably, a previous study conducted among middle-aged and older Taiwanese adults identified two major dietary patterns: a vegetable–fruit pattern and a meat–processed food pattern [19]. When compared with the present findings, our TC1 closely resembles the vegetable–fruit dietary pattern, as both are characterized by high intakes of fruits, vegetables, and abundant antioxidant nutrients. Similarly, TC2 corresponds to the meat–processed food dietary pattern, which is likewise dominated by meat and high-fat food consumption. Furthermore, TC3 may reflect the typical high-carbohydrate dietary pattern prevalent among Taiwanese older adults, which is largely driven by staple foods such as rice, noodles, and congee. Taken together, these comparisons highlight the existence of distinct and consistent dietary structures within the Taiwanese older adult population.

In the longitudinal analysis, PC1, which accounted for the largest proportion of variance (60.9%) (Appendix A), demonstrated a significant protective effect only among APOE4 carriers. This finding is consistent with previous studies showing that plant-based diets rich in dietary fiber and antioxidants are generally associated with a lower risk of cognitive decline [20,21,22,23,24]. Notably, in our Taiwanese cohort, this protective effect was evident exclusively in individuals carrying the APOE4 allele. The underlying mechanisms likely relate to the greater susceptibility of APOE4 carriers to chronic inflammation and oxidative stress [25,26,27,28]. The protective nutrients in TC1 may exert their benefits through multiple pathways: Dietary fiber modulates gut microbiota and promotes short-chain fatty acid production to reduce systemic inflammation [29,30]; β-carotene, vitamin A, and vitamin C act as antioxidants that neutralize free radicals and mitigate neuronal damage [31,32,33]; and folate supports neural health by regulating homocysteine metabolism and DNA methylation [34,35,36]. Recent studies further report that high dietary fiber intake confers notably strong cognitive protection in APOE4 carriers [37] supporting our observations (Figure 3).

In contrast, higher TC2 scores were associated with a faster rate of cognitive decline among APOE4 carriers (Figure 3). This observation is consistent with evidence that APOE4 carriers are more vulnerable to dysregulation of lipid metabolism [38,39,40]. Because TC2 primarily reflects higher intakes of fatty acids, cholesterol, and protein, representing a more animal-based, high-fat dietary pattern, elevated TC2 may exacerbate lipid imbalance, promote chronic low-grade inflammation, and increase cardiovascular risk [41,42]. These interconnected metabolic and inflammatory pathways may partly explain the adverse association between TC2 and cognitive decline, and why it appears more pronounced in APOE4 carriers. Supporting this interpretation, our supplementary analyses showed that TC2 was significantly correlated with circulating LDL cholesterol and total cholesterol (Appendix A), further suggesting a link to lipid metabolic disruption.

The model in this study distinguished between baseline diagnoses and new-onset conditions to clarify how pre-existing versus emerging health problems relate to cognitive change (Table 2). Baseline diseases reflect long-standing vascular or metabolic burden, whereas new-onset diagnoses capture more recent changes in health status during follow-up (Table 2). In our analyses, hypertension and hyperlipidemia did not show significant associations in either baseline or new-onset models. In contrast, diabetes showed clearer effects (Table 2). Baseline diabetes was significantly associated with cognitive decline (Table 2), suggesting that long-term metabolic burden may exert cumulative and sustained effects on cognitive trajectories. New-onset diabetes showed similar directional patterns (Table 2), although the associations were not statistically significant, indicating that recent changes in metabolic health may also contribute to cognitive vulnerability, but to a smaller degree. Together, these findings highlight the temporal dimension of disease-related risk and the importance of monitoring both chronic and newly diagnosed comorbidities when evaluating cognitive health.

In the genotype-stratified analysis, the interaction between TC2 and APOE status revealed meaningful heterogeneity in the direction of associations. Among APOE4 carriers, higher TC2 scores were associated with poorer cognitive performance, whereas the non-E4 group (APOE2/3) showed an opposite directional trend, although the slope did not reach statistical significance. This “opposite-direction but non-significant” pattern suggests that the biological interpretation of TC2 may differ by APOE genotype. In non-E4 individuals, higher TC2 may partially reflect a more favorable lipid milieu, potentially involving protective lipid components such as omega-3 fatty acids, a pattern consistent with Western literature linking omega-3–related lipids to cognitive benefits [43,44,45]. In contrast, in the APOE4 context, where protective lipid components may exert weaker effects, the detrimental features of elevated lipids, such as metabolic stress and low-grade chronic inflammation, may predominate and contribute to cognitive decline.

Age and education remain key determinants of cognitive function in later life [46] and our results likewise show that age and years of education are closely associated with cognitive change (Table 2). The APOE4 × TC1 interaction was associated with a 0.115-unit slower annual rate of cognitive decline per one-unit increase in TC1. Within the linear model, this magnitude is roughly comparable to the effect of 3.96 additional years of education (education coefficient = 0.029) or being approximately 6.39 years younger (age coefficient = −0.018). Although aging and educational attainment are not readily modifiable in later life, these findings suggest that optimizing dietary patterns and managing chronic conditions may help mitigate cognitive decline, particularly among individuals with higher genetic risk.

In this study, the modest model fit (R^2^ ≈ 0.10) should be interpreted in light of the outcome and study design. Longitudinal cognitive aging research indicates that inter-individual differences in cognitive change lack strong single predictors, and baseline demographic and genetic factors typically explain only a limited proportion of variance in decline rates [47,48]. Moreover, ΔMMSE is a bounded difference score and is susceptible to measurement error, ceiling effects, and practice effects, which reduce the reliability of observed change and constrain the maximum explainable variance, leading to attenuation of associations [49,50]. Because only two waves of follow-up were available, ΔMMSE represents a single linear difference and does not allow stable estimation of individual slopes or non-linear decline, further contributing to modest R^2^ values even when predictors are etiologically relevant [51]. Importantly, the aim of this study is explanatory rather than purely predictive, and methodological literature emphasizes that there is no universal R^2^ cutoff in clinical or behavioral research; relatively low R^2^ values can still be meaningful when effects are robust and interpretable [52,53].

While this study provides novel insights, several limitations should be acknowledged. First, as an observational study, our findings remain vulnerable to residual confounding. Although we adjusted for multiple important covariates, factors such as physical activity, socioeconomic status, medication use, depressive symptoms, and broader gut health-related conditions may not have been fully captured or measured with sufficient precision. Unmeasured or imperfectly measured confounders could therefore bias the observed associations. Future studies incorporating more comprehensive assessments of these factors, as well as experimental or quasi-experimental designs, are warranted to strengthen causal inference. Second, dietary intake was self-reported and may be affected by recall bias and social desirability bias. To assess the plausibility of FFQ-derived nutrient estimates, we performed linear regression analyses comparing estimated nutrient intake with blood biochemical markers, and observed directionally consistent associations for selected indicators, supporting that the FFQ can reflect relative intake differences at the population level (Appendix A). However, the FFQ primarily captures relative intake and dietary patterns, and is not intended to provide precise quantitative estimates for individual-level assessment or clinical interpretation. Moreover, self-report error is often non-differential, which would be expected to attenuate associations toward the null and render our estimates conservative. Third, because of the observational design, causal relationships cannot be directly inferred, and residual confounding cannot be completely ruled out. Fourth, cognitive function was assessed using the MMSE, which measures global cognition but may be less sensitive to subtle or domain-specific changes. More sensitive instruments such as the MoCA and neuroimaging biomarkers were not available in the current dataset and therefore could not be evaluated. In addition, the two-wave follow-up design enabled within-subject comparisons but offered less temporal resolution than multi-wave longitudinal data. Participants who completed both waves differed from those with baseline-only data in sociodemographic characteristics, comorbidities, and baseline MMSE scores, suggesting potential attrition-related selection bias. With cognitive function assessed at only two time points, we were also limited in our ability to model individual cognitive trajectories, detect potential non-linear change, or obtain more precise estimates of cognitive decline; additional follow-up waves will be essential to validate and extend the longitudinal patterns observed here. Finally, individuals carrying the ε2/ε4 genotype were excluded from the analyses due to their small sample size (*n* = 19), which limited our ability to evaluate cognitive associations in this subgroup. Additional follow-up waves and larger samples will be essential to validate and extend the longitudinal patterns observed here.

To our knowledge, this study is among the first longitudinal investigations in an East Asian population to examine how changes in nutrient-derived dietary patterns relate to cognitive decline, with particular attention to the APOE4 genotype. Our findings indicate that increases in plant-based, antioxidant-rich dietary patterns (TC1) were associated with a slower rate of cognitive decline, whereas shifts toward animal- and fat-rich dietary patterns (TC2) were linked to accelerated decline. Importantly, these associations were observed exclusively among APOE4 carriers, suggesting heightened genetic sensitivity to dietary influences on cognitive aging. These results underscore the critical role of APOE genotype in modulating the relationship between diet and cognition and support the existence of genotype-specific biological pathways that shape cognitive trajectories.

## 5. Conclusions

These findings raise the possibility that precision nutrition approaches tailored to genetic risk profiles warrant further investigation, given the limitations discussed. For older adults carrying the APOE4 allele, promoting plant-based dietary patterns while limiting animal- and fat-rich foods may represent a promising strategy to mitigate cognitive decline, as APOE4 carriers are both a high-risk group for cognitive deterioration and appear particularly responsive to dietary modification. Targeted nutritional interventions may therefore provide meaningful benefits for maintaining cognitive health, and future interventional studies are needed to validate these observations and elucidate the underlying causal mechanisms.

## Figures and Tables

**Figure 1 nutrients-18-00106-f001:**
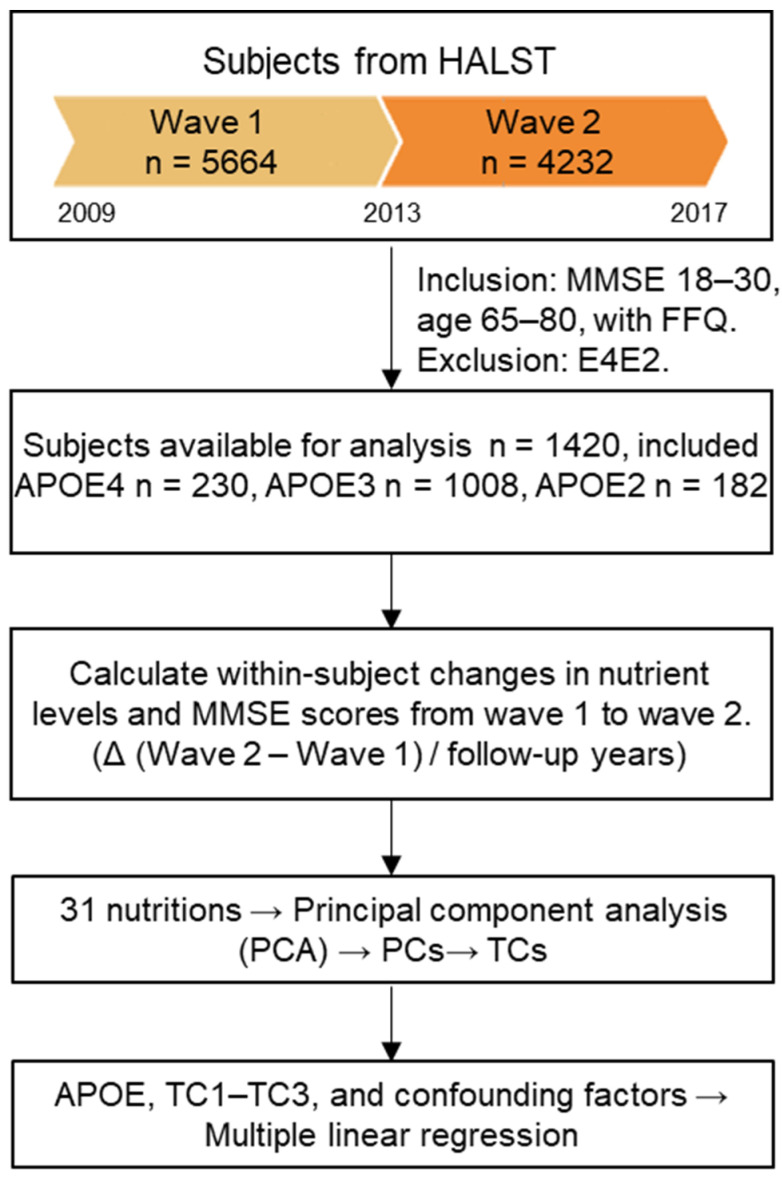
Flowchart of study design and subject selection. Participants were drawn from Wave 1 (*n* = 5664; initiated in 2009) and Wave 2 (*n* = 4232; initiated in 2013) of the HALST cohort. After applying exclusion criteria and stratifying by APOE genotype, 1420 subjects were included in the longitudinal model (Δ dietary intake vs. Δ MMSE). Nutrient-based dietary patterns were derived using principal component analysis (PCA) to obtain principal components (PCs), and oblimin-derived pattern scores were used to generate the dietary pattern change variables (TCs). Multiple linear regression models were performed after adjusting for APOE, age, sex, education, energy intake, TCs, and the three major chronic diseases—hypertension, diabetes, and hyperlipidemia.

**Figure 2 nutrients-18-00106-f002:**
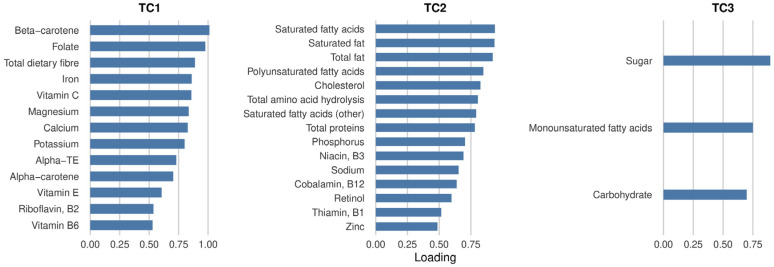
Principal component analysis (PCA) loadings of 31 dietary nutrients. Three components were retained based on the screen plot with oblimin-derived scores. Higher loading values indicate greater contribution of the nutrient to the dietary pattern represented by each transformed component (TC).

**Figure 3 nutrients-18-00106-f003:**
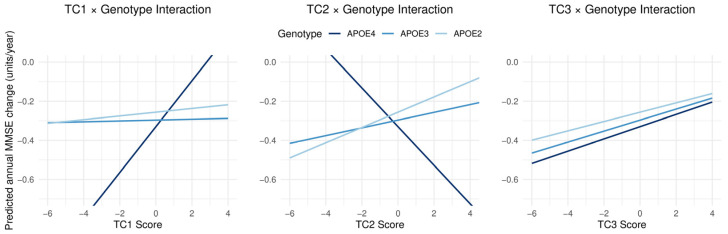
Interactions between nutrient-derived transformed components (TCs) and APOE genotype on cognitive performance. Predicted MMSE scores are plotted against TC1, TC2, and TC3 scores, stratified by APOE genotype (E2, E3, E4) with hypertension, diabetes, and hyperlipidemia. Significant interactions were observed for TC1 × genotype and TC2 × genotype, indicating differential associations between nutrient patterns and cognition across APOE groups, whereas TC3 × genotype showed no meaningful interaction.

**Table 1 nutrients-18-00106-t001:** Baseline characteristics.

Characteristic	APOE4 (*n* = 230)	APOE3 (*n* = 1008)	APOE2 (*n* = 182)	*p*-Value
Age (Mean ± SD)	70.31 ± 3.54	70.45 ± 3.57	70.73 ± 3.52	0.480
Female (%)	132 (57.4)	550 (54.6)	101 (55.5)	0.735
Education (year)	8.34 ± 4.39	8.21 ± 4.53	7.90 ± 4.51	0.594
ΔMMSE	−0.31 ± 0.56	−0.29 ± 0.52	−0.27 ± 0.56	0.701
ΔTotal energy intake (kcal)	−24.24 ± 134.82	−10.48 ± 123.54	−11.39 ± 125.51	0.322
ΔTotal carbohydrate (g)	−6.00 ± 23.63	−2.80 ± 20.68	−2.56 ± 22.05	0.108
ΔTotal dietary fiber (g)	−0.62 ± 2.32	−0.47 ± 2.32	−0.44 ± 2.03	0.637
ΔTotal proteins (g)	−1.08 ± 6.68	−0.67 ± 6.38	−0.64 ± 5.61	0.659
ΔTotal fat (g)	0.41 ± 4.57	0.33 ± 4.42	0.17 ± 5.10	0.860

Footnote: Age and education (years) were measured at baseline (Wave 1). Within-subject changes in nutrient levels and Mini-Mental State Examination (MMSE) scores were calculated as Δ (Wave 2–Wave 1) per follow-up year.

**Table 2 nutrients-18-00106-t002:** Multiple linear regression model examining the associations of APOE and dietary patterns with cognitive function with 3 chronic diseases.

Variable	Estimate	Std. Error	95% CI	*p*-Value
Intercept	0.751	0.284	(0.195, 1.307)	0.008 **
Genotype				
APOE4 carrier (vs E3)	−0.033	0.037	(−0.106, 0.039)	0.369
APOE2 carrier (vs E3)	0.040	0.041	(−0.040, 0.121)	0.323
Age	−0.018	0.004	(−0.025, −0.010)	<0.001 ***
Education	0.029	0.003	(0.023, 0.035)	<0.001 ***
Sex (female)	−0.006	0.029	(−0.063, 0.050)	0.831
ΔTotal energy intake	0.000	0.000	(−0.001, 0.000)	0.112
TC1	0.002	0.020	(−0.036, 0.040)	0.911
TC2	0.020	0.024	(−0.027, 0.066)	0.403
TC3	0.028	0.024	(−0.019, 0.075)	0.239
Hypertension (ref = No diagnosis)				
Baseline diagnosis	−0.043	0.030	(−0.102, 0.015)	0.145
New-onset by wave 2	−0.050	0.045	(−0.139, 0.038)	0.265
Diabetes (ref = No diagnosis)				
Baseline diagnosis	−0.094	0.035	(−0.163, −0.025)	0.008 **
New-onset by wave 2	−0.094	0.051	(−0.194, 0.006)	0.065 .
Hyperlipidemia (ref = No diagnosis)				
Baseline diagnosis	0.050	0.033	(−0.014, 0.114)	0.127
New-onset by wave 2	0.036	0.041	(−0.044, 0.116)	0.379

Footnote: Summary statistics of the multiple linear regression model. Std. Error: Standard Error. 95% CI: 95% Confidence interval. Residual standard error: 0.502 on 1398 degrees of freedom. Multiple R-squared: 0.108. Adjusted R-squared: 0.095. F-statistic: 8.068 on 21 and 1398 DF, *p*-value: < 2.2 × 10^−16^. Significance codes: *** < 0.001; ** < 0.01.

**Table 3 nutrients-18-00106-t003:** Multiple linear regression model examining the associations of APOE and transformed components TC1–3.

Interaction Terms	Estimate	Std. Error	95% CI	*p*-Value	*q*-Value
APOE4 × TC1	0.115	0.044	(0.029, 0.201)	0.009 **	0.026 *
APOE2 × TC1	0.007	0.050	(−0.091, 0.106)	0.884	0.939
APOE4 × TC2	−0.119	0.043	(−0.202, −0.035)	0.005 **	0.026 *
APOE2 × TC2	0.019	0.046	(−0.072, 0.110)	0.678	0.939
APOE4 × TC3	0.003	0.042	(−0.080, 0.086)	0.939	0.939
APOE2 × TC3	−0.004	0.043	(−0.088, 0.079)	0.920	0.939

Footnote: Summary statistics of the multiple linear regression model. Std. Error: Standard Error. 95% CI: 95% Confidence interval. Residual standard error: 0.502 on 1398 degrees of freedom. Multiple R-squared: 0.108. Adjusted R-squared: 0.095. F-statistic: 8.068 on 21 and 1398 DF, *p*-value: < 2.2 × 10^−16^. Significance codes: ** < 0.01; * < 0.05; *q*-value: adjust by FDR correction.

## Data Availability

The data presented in this study are available on request from the corresponding author due to privacy, ethical, and regulatory restrictions, and access may require additional approvals.

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
