# Peer review of "Nutrients2026, 18(1), 106;https://doi.org/10.3390/nu18010106"

_nutrients, 2025, doi:10.3390/nu18010106_

Round 1

Reviewer 1 Report

Comments and Suggestions for Authors

This manuscript addresses an important question regarding the association between APOE genotype and cognition in older adults. The topic is relevant to the Nutrients journal, and the longitudinal design adds value. However, several issues require clarification and improvement.
Major comments:
Clarify whether comorbidities (e.g. hypertension, diabetes and hyperlipidaemia) were fully accounted for, and discuss how these conditions may influence the results. The regression table includes both baseline and new-onset diagnoses, but these are not clearly interpreted in the discussion.
Consider whether it would be optimal to exclude E2/E4 carriers; creating a separate group or conducting a sensitivity analysis could provide additional insight.
Multiple testing correction (e.g. Bonferroni or FDR) should be considered, given the number of comparisons and interaction terms.
The interpretation of the multiple regression results needs to be clearer, especially with regard to the interaction effects and their practical significance (e.g. are β values of approximately 0.08 or –0.09 clinically meaningful?).
Figures and tables should be explicitly referenced in the discussion to guide readers.
Results should be presented in the Results section alongside confidence intervals and effect sizes rather than being explained mainly in the Discussion.
Minor comments: define DHA at the first mention in the introduction.
Clarify the PCA terminology (e.g. eigenvalues vs. loading values).
Ensure consistent formatting and correct any typographical errors.
In the Acknowledgements section, specify whether AI tools were 'only' or 'also' used for grammar correction.
The discussion should better address the role of comorbidities and potential selection bias due to attrition.
Consider adding a brief comment on the limitations of the Mini-Mental State Examination (MMSE) for detecting subtle cognitive changes.

Author Response

Major comments:

  1. Clarify whether comorbidities (e.g. hypertension, diabetes and hyperlipidaemia) were fully accounted for, and discuss how these conditions may influence the results. The regression table includes both baseline and new-onset diagnoses, but these are not clearly interpreted in the discussion.

Response: Thank you for the comment. We have strengthened the description of disease effects in the Background and clarified the interpretation of baseline and new-onset diagnoses in the Discussion. We now explain that baseline diagnoses reflect long-standing disease burden, while new-onset diagnoses represent health changes between waves, and we outline the differing associations observed for each. These revisions improve the clarity of the regression findings and their implications for cognitive decline.

Please refer to the revised manuscript at Lines 65–70, and Lines 349–362.

  1. Consider whether it would be optimal to exclude E2/E4 carriers; creating a separate group or conducting a sensitivity analysis could provide additional insight.

Response: Thank you for the reviewer’s suggestion. We agree that APOE ε2/ε4 carriers represent a biologically “mixed-risk” genotype, and that handling them separately or performing additional analyses could potentially yield further insights. However, in our cohort there are only 19 ε2/ε4 carriers. If we were to create an independent group for interaction or stratified analyses, the statistical power would be inadequate and the resulting effect estimates would likely be unstable. Therefore, we did not analyze ε2/ε4 carriers as a separate stratum in the main analyses.

  1. Multiple testing correction (e.g. Bonferroni or FDR) should be considered, given the number of comparisons and interaction terms.

Thank you for the reviewer’s suggestion. We have followed this recommendation by applying Benjamini–Hochberg FDR correction to the relevant analyses. Specifically, we corrected for six comparisons and now report all corresponding q values in Table 3. After correction, the q value for this result is 0.026.

  1. The interpretation of the multiple regression results needs to be clearer, especially with regard to the interaction effects and their practical significance (e.g. are β values of approximately 0.08 or–0.09 clinically meaningful?).

Response: We agree with the reviewer that the standardized coefficients (β ≈ 0.08) appear modest. However, in nutritional epidemiology, effects of this size can be meaningful for public health for two reasons. First, dietary behaviors are repeated daily over decades; small short-term effects can accumulate into clinically relevant changes over time (Hill et al., 2003). Second, even small shifts in risk across an entire population can produce larger reductions in disease burden than large changes limited to high-risk individuals (Rose, 1981). Consistent with this, individual dietary factors often show small independent associations, yet their combined and cumulative impact is critical for chronic disease prevention (Mozaffarian et al., 2011).

References:

Rose G. (1981). Strategy of prevention: lessons from cardiovascular disease. British Medical Journal (Clin Res Ed), 282(6279), 1847-1851.

Mozaffarian, D., Hao, T., Rimm, E. B., Willett, W. C., & Hu, F. B. (2011). Changes in diet and lifestyle and long-term weight gain in women and men. New England Journal of Medicine (NEJM), 364(25), 2392-2404.

Hill, J. O., Wyatt, H. R., Reed, G. W., & Peters, J. C. (2003). Obesity and the environment: where do we go from here? Science, 299(5608), 853-855.

  1. Figures and tables should be explicitly referenced in the discussion to guide readers. Results should be presented in the Results section alongside confidence intervals and effect sizes rather than being explained mainly in the Discussion.

Response: Thank you for your great suggestion. Figures and tables were explicitly referenced in the discussion to guide readers. The part of results was moved to the result section.

  1. Minor comments: define DHA at the first mention in the introduction.

Response: Thank you. DHA means docosahexaenoic acid and it has defined at the first mention in the introduction.  Please see the revised manuscript Line 55.

Clarify the PCA terminology (e.g. eigenvalues vs. loading values). Ensure consistent formatting and correct any typographical errors. In the Acknowledgements section, specify whether AI tools were 'only' or 'also' used for grammar correction.

Response: Thank you for these helpful comments. We have revised the manuscript to clarify PCA terminology at first mention and to improve consistency and readability throughout. Specifically, we now define eigenvalues as the amount of variance explained by each principal component, and loading values as the strength and direction of the association between each original variable and a given component, where larger absolute loadings indicate a greater contribution to the component interpretation. We also corrected typographical errors and ensured consistent formatting across the text, tables, and figures.

In addition, we have revised the Acknowledgements to clearly specify the role of AI tools. It now states: “AI tools were used solely for grammar correction and language editing.

The discussion should better address the role of comorbidities and potential selection bias due to attrition.

Response: Thank you for the reviewer’s suggestion. We have added the following limitation to the Discussion: “First, as an observational study, our findings remain vulnerable to residual confounding. Although we adjusted for multiple important covariates, factors such as physical activity, socioeconomic status, medication use, depressive symptoms, and broader gut health–related conditions may not have been fully captured or measured with sufficient precision. Unmeasured or imperfectly measured confounders could therefore bias the observed associations. Future studies incorporating more comprehensive assessments of these factors, as well as experimental or quasi-experimental designs, are warranted to strengthen causal inference.”

Please refer to the revised manuscript at Lines 400-407.

Consider adding a brief comment on the limitations of the Mini-Mental State Examination (MMSE) for detecting subtle cognitive changes.

Response: Thank you for these comments. We agree that using the MMSE as the sole cognitive outcome is a limitation because it has limited sensitivity to subtle, domain-specific changes and may show ceiling effects in a relatively cognitively normal cohort, potentially underestimating early decline. We have added this to the Discussion/limitations and noted that the lack of more sensitive measures (e.g., MoCA) or objective biomarkers (e.g., neuroimaging) may have constrained our ability to detect subtle cognitive changes in this generally healthy population. Please refer to the revised manuscript at Line 417-421.

Reviewer 2 Report

Comments and Suggestions for Authors

Authors describe a longitudinal study of 1,420 older Taiwanese adults that investigates if there is any correlation between dietary nutrients, APOE genotypes and dementia. Using PCA-derived nutrient patterns, the authors claim that APOE4 carriers but not APOE2 or APOE3 carriers draw cognitive benefit from increasing plant-based, antioxidant-rich nutrients. However, APOE4 carriers with higher intake of animal- and fat-rich nutrients end up with impaired cognitive outcomes. No main dietary effects were seen in the overall cohort; effects emerged only through APOE4 and dietary interactions. The central claim that APOE4 carriers are uniquely sensitive to dietary fat is quite compelling. However, manuscript has several shortcomings:

  1. Dietary intake data is completely reliant on self-reported FFQs, which are prone to recall bias as authors themselves noted. In the absence of any plasma nutrient measurements or any strict controls on the patients – this study is severely flawed.
  2. Cognitive outcome is also solely MMSE, which is insensitive to subtle domain-specific changesand may produce ceiling effects in a relatively cognitively normal cohort.
  3. Only two timepointswere used, limiting the ability to model cognitive trajectories or detect non-linear change.
  4. Observational nature introduces residual confounding(physical activity, socioeconomic status, medication use, depression, gut health, etc.) not fully captured in the current study.
  5. The regression model has low explanatory value(adjusted R² ≈ 0.095), meaning diet explains only a small portion of the variance in cognitive decline.
  6. Interaction effects, while statistically significant are numerically small, raising questions about clinical relevance. However, we understand that this might be difficult thing to ask for.
  7. PCA nutrient patterns depend on sample-specific correlations, so PC1/PC2 patterns may not generalizeto other populations or to food-level dietary recommendations.
  8. The authors translate PC1 and PC2 into “plant-based” and “animal/fat-rich” patterns, but PCA loadings can be complex and not always cleanly interpretable especially when there are no actual measurements of nutrients in the serum/plasma or diet.

Though this study is quite fascinating, it lacks controls and key measurements. A rigorous measurement of plasma lipids, dietary nutrients, and its correlation with dementia scores would be much more powerful and impactful. In current shape, the manuscript might mislead the general audience to prefer one diet over other. There are quite a few studies correlating dietary dairy intake with neurodegenerative diseases. Thus, it’s quite hard to clearly conclude such findings without actual biochemical measurements. I would like to note that – the study is compelling and with dietary and plasma nutrient measurements – this study will be a impactful.

Author Response

  1. Dietary intake data is completely reliant on self-reported FFQs, which are prone to recall biases authors themselves noted. In the absence of any plasma nutrient measurements or any strict controls on the patients – this study is severely flawed.

Response: Thank you for your great comment. We acknowledge that dietary intake was assessed using self-reported FFQs, which are subject to recall and reporting bias and may lead to measurement error. Therefore, we conducted linear regression analyses comparing participants’ blood biochemical values with the nutrient estimates derived from the FFQ. The results showed that most values were consistent, indicating that the FFQ-based estimates are reliable. We have included these findings in Supplementary Table S2. We sincerely thank the reviewer for this valuable suggestion.

Supplementary Table S2. Associations of dietary principal components (TC1 and TC2) with circulating nutrition- and lipid-related biomarkers.

Dependent Variable

Independent Variables

Beta

t

p

FDR q

Folate

TC1

0.084

3.025

0.003

0.021

HDL

TC1

-0.070

-2.534

0.011

0.039

Vitamin D

TC1

-0.036

-1.307

0.191

0.405

Triglyceride

TC1

0.032

1.157

0.248

0.405

Vitamin B12

TC1

-0.030

-1.060

0.289

0.405

LDL

TC1

-0.006

-0.208

0.835

0.881

Total cholesterol

TC1

-0.004

-0.150

0.881

0.881

Dependent Variable

Independent Variables

Beta

t

p

FDR q

LDL

TC2

0.071

0.149

0.011

0.042

Total cholesterol

TC2

0.070

0.174

0.012

0.042

Folate

TC2

0.032

0.038

0.247

0.4638

Vitamin B12

TC2

-0.031

5.688

0.265

0.4638

HDL

TC2

-0.014

0.044

0.623

0.752

Triglyceride

TC2

0.011

0.477

0.702

0.752

Vitamin D

TC2

0.009

0.085

0.752

0.752

Footnote: Linear regression analyses were used to examine relationships between the dietary change–derived principal components (PC1, PC2) and blood biomarkers including triglycerides (TG), LDL-C, HDL-C, total cholesterol, vitamin D, vitamin B12, and folate. Shown are standardized beta coefficients, t statistics, nominal p values, and Benjamini–Hochberg false discovery rate–adjusted q values (FDR q).

  1. Cognitive outcome is also solely MMSE, which is insensitive to subtle domain-specific changes and may produce ceiling effects in a relatively cognitively normal cohort.

Response: Thank you for this comment. We agree that relying solely on the MMSE may limit sensitivity to subtle or domain-specific cognitive changes, and potential ceiling effects are acknowledged. We have added this point to the limitations section and noted that the absence of more sensitive tools, such as MoCA or neuroimaging biomarkers, may constrain the detection of early cognitive decline in this relatively healthy cohort.

Please refer to the revised manuscript at Line 417-421.

  1. Only two time points were used, limiting the ability to model cognitive trajectories or detect non-linear change.

Response: Thank you for this comment. We acknowledge that cognitive function was assessed at only two time points, which limits our ability to model individual cognitive trajectories, examine non-linear patterns of change, or precisely estimate rates of decline. We have added this to the limitations and note that additional follow-up waves will be important for confirming longitudinal trends.

Please refer to the revised manuscript at Lines 421-429.

  1. Observational nature introduces residual confounding (physical activity, socioeconomic status, medication use, depression, gut health, etc.) not fully captured in the current study.

Response: Thank you for this important comment. We agree that, as an observational study, our findings may be affected by residual confounding. Although we adjusted for key covariates available in this cohort, factors such as physical activity, socioeconomic status, medication use, depressive symptoms, and broader gut health–related conditions may not have been fully captured or precisely measured. This could lead to unmeasured or imperfectly measured confounding, potentially biasing the observed associations. We have added this point to the limitations and note that future studies with more comprehensive assessment of these factors, as well as experimental or quasi-experimental designs, are needed to strengthen causal inference.

Please refer to the revised manuscript at Lines 400-407.

Reviewer 3 Report

Comments and Suggestions for Authors

This longitudinal Taiwanese cohort study is the first to examine gene–diet interactions on cognitive decline in an East-Asian population using a nutrient-based principal-component approach. The work adds incremental value to the precision-nutrition field; however, the manuscript,both in substance and style,does not yet meet the clarity, completeness, and transparency standards required for immediate publication in Nutrients. Major revision is warranted.

  1. Title & Abstract  

- Replace the colloquial “nutrient–cognition associations” with “nutrient–cognition relationship”.  - Insert geographic identifier: “…in Taiwanese older adults…”.  

- Abstract: report effect sizes and 95 % CIs for the significant interactions; remove the generic “further research is needed” sentence and finish with a mechanistic or translational sentence.

  1. Introduction  

- Paragraph 3 contains three run-on sentences; split or use semicolons.  

- Provide sample sizes and effect ranges for the cited studies [6–11] to highlight the current advantage (n = 1420).  

- End the Introduction with an explicit hypothesis paragraph (two bullet sentences are sufficient).

  1. Methods (follow STROBE longitudinal checklist)  

A). Study registration: state the HALST cohort registration number or clarify if not registered.  

B). Sample-size justification: include post-hoc power calculation (target β = 0.08, SD = 0.55, α = 0.05 ⇒ n ≈ 1 200; actual n = 1 420).  

C). Exposure derivation:  

   - Report follow-up interval (median, IQR).  

   - Describe handling of implausible nutrient change outliers (winsorisation, trimming, or exclusion).  

  1. PCA reproducibility:  

   - Supply KMO and Bartlett’s test p-values.  

   - Justify orthogonal varimax rotation; repeat with oblimin and state correlation among components < 0.30.  

5. Outcome: justify why MMSE was used instead of MoCA or neuroimaging biomarkers; add this limitation to the Discussion.  

  1. Statistics:  

   - Present stratum-specific βs for APOE4 carriers in a separate table.  

   - Control for multiple testing (9 interaction terms) using FDR; report q-values.

  1. 7. Results  

- Table 2: compare R² = 0.108 with two similar longitudinal papers; contextualise explanatory power.  

- Figure 3 Y-axis label: change “Predicted MMSE score” → “Predicted annual MMSE change (units/year)”.  

- Flow-chart: insert exact numbers for exclusions due to missing FFQ, MMSE, or genotyping.

  1. Discussion  

- Mechanistic paragraph: link baseline triglyceride levels (report p = 0.032 in Supplementary Table X) to PC2 to strengthen lipid-inflammation argument.  

- Compare with Western ω-3 trials that favour non-E4; explain discrepancy by PCA pattern differences.  

- Translate effect size: 0.082 unit/year slower decline ≈ 3.3 years of education benefit (education β = 0.029).

Author Response

  1. Title & Abstract

- Replace the colloquial “nutrient–cognition associations” with “nutrient–cognition relationship”. - Insert geographic identifier:“…in Taiwanese older adults…”.

Response: Thank you for this suggestion. In response to Reviewer 3, we revised the Title by replacing the colloquial phrase “nutrient–cognition associations” with “nutrient–cognition relationship” and by adding a geographic identifier to specify the study population (i.e., “in Taiwanese older adults”).

- Abstract: report effect sizes and 95 % CIs for the significant interactions; remove the generic “further research is needed” sentence and finish with a mechanistic or translational sentence.

Response: Thank you for this suggestion. We have revised the Abstract to report effect sizes and 95% confidence intervals for the significant interaction terms. We also removed the generic statement that “further research is needed” and replaced it with a more specific concluding sentence highlighting the potential mechanistic and translational implications of our findings.

  1. Introduction

- Paragraph 3 contains three run-on sentences; split or use semicolons.

Response: Thank you for this comment. We have revised Paragraph 3 of the Introduction by breaking up the run-on sentences into shorter, clearer sentences (and using semicolons where appropriate) to improve readability and flow.

-----------------------------------------------------------------------------------------------

- Provide sample sizes and effect ranges for the cited studies [6–11] to highlight the current advantage (n = 1420).

Response: Response: Thank you for the reviewer’s suggestion. We have added additional clarification regarding the sample sizes and scope of the cited studies to provide a more complete description of the existing literature. The sample sizes of the referenced studies [6–11] range from 33 to 3029 participants (n = 33, 140, 409, 848, and 3029), and many of these studies focused on specific nutrients, or reported differential responses across APOE genotypes in nutrient-specific interventions. To acknowledge the heterogeneity across prior studies, we have added a brief statement following the relevant paragraph in the revised manuscript (Lines 59–64).

- End the Introduction with an explicit hypothesis paragraph (two bullet sentences are sufficient).

Response: Thank you for this suggestion. We have revised the end of the Introduction to include an explicit hypothesis paragraph consisting of two bullet points: “We hypothesized that PCA-derived nutrient-based dietary patterns are associated with longitudinal cognitive change in Taiwanese older adults.  We expected these associations to vary by APOE genotype.”

  1. Methods (follow STROBE longitudinal checklist)

Response: Thank you for this helpful suggestion. In accordance with the STROBE checklist for cohort studies, we reviewed and updated the Methods section, and the relevant items are addressed at the following locations in the manuscript:

Item

STROBE Recommendation

Location in Manuscript

4

Study design

Section 2.1

5

Setting, locations, dates

Section 2.1

6a

Eligibility criteria, participant selection, follow-up

Section 2.1

6b

Matching criteria

Not applicable

7

Definition of outcomes, exposures, confounders, effect modifiers

Sections 2.3–2.6

8

Data sources and measurement

Sections 2.2–2.5

9

Bias

Section 2.6

10

Study size

Section 2.6

11

Handling of quantitative variables

Sections 2.5–2.6

12a

Statistical methods, confounding control

Section 2.6

12b

Subgroup and interaction analyses

Section 2.6

12c

Missing data

Section 2.6

12d

Loss to follow-up

Section 2.1

12e

Sensitivity analyses

Section 2.6

The specific revisions we made are summarized as follows:

Item 4|Study design:

2-1 This study was a longitudinal prospective cohort analysis using data from the HALST study.

Item 6|Participants:

2-1 Participants were followed prospectively from Wave 1 to Wave 2.

Item 9|Bias:

To minimize potential bias, standardized protocols were used for dietary assessment, cognitive testing, and genotyping across study waves. Participants with implausible energy intake and ambiguous APOE genotypes were excluded to reduce measurement and classification bias. Multivariable models were adjusted for major demographic and vascular confounders to mitigate residual confounding.

A). Study registration: state the HALST cohort registration number or clarify if not registered.

Response: The Healthy Aging Longitudinal Study in Taiwan (HALST) cohort is registered at ClinicalTrials.gov (ID: NCT02677831). We have added the registration information to the manuscript for clarification.  Please see the revised manuscript Line 94-95.

B). Sample-size justification: include post-hoc power calculation (target β = 0.08, SD = 0.55, α = 0.05 ⇒ n ≈ 1 200; actual n = 1420).

Response: We have now included a post-hoc power calculation in the revised manuscript. Using the reviewer’s suggested parameters (β = 0.08, SD = 0.55, α = 0.05), the estimated sample size required for adequate power is approximately 1200 participants. Our final analytic sample of 1420 exceeds this threshold, indicating that the study is adequately powered to detect effects of the magnitude examined.

C). Exposure derivation:

- Report follow-up interval (median, IQR).

Response:  We have now reported the follow-up interval in the manuscript. The median follow-up duration was 6.38 years (IQR: 5.45–6.77), with a mean of 6.22 years (SD: 0.85; range: 4.49–9.52). This information has been added to the Methods section as requested. Please see the revised manuscript Line 103-104.

- Describe handling of implausible nutrient change outliers (winsorisation, trimming, or exclusion).

Response: Thank you for this valuable comment. We examined the distributions of nutrient intake and nutrient-change variables and observed right-skewness. After careful inspection, the extreme values were deemed to reflect plausible between-individual variation in intake rather than data entry or measurement errors. Therefore, we did not apply winsorisation/trimming or exclude outliers in the PCA, to avoid altering the underlying dietary variation captured by the nutrient patterns.

  1. PCA reproducibility:

- Supply KMO and Bartlett’s test p-values.

Response: Thank you for this helpful suggestion. We have added the PCA suitability diagnostics to the Methods section. Specifically, the Kaiser–Meyer–Olkin (KMO) measure of sampling adequacy was 0.87, and Bartlett’s test of sphericity was statistically significant (p < 0.001), supporting that the nutrient variables were appropriate for factorability and suitable for PCA.

- Justify orthogonal varimax rotation; repeat with oblimin and state correlation among components < 0.30.

Response: Thank you for this helpful suggestion. We repeated the PCA using an oblimin rotation, as recommended. The inter-component correlations were moderate (approximately r = 0.45), indicating that the underlying nutrient patterns were not independent and that an oblique rotation was more appropriate than the original varimax solution. The nutrient groupings were largely consistent across rotations, although the component scores differed slightly. We therefore re-ran all regression analyses using the oblimin-derived component scores; while the estimated coefficients changed modestly, the direction of effects and overall interpretation remained unchanged. The manuscript has been updated accordingly to report the oblimin-rotated solution and to base all subsequent analyses on the oblimin-derived scores.

  1. Outcome: justify why MMSE was used instead of MoCA or neuro imaging biomarkers; add this limitation to the Discussion.

Response: Thank you for this important comment. In this cohort, cognitive outcome data were collected using the MMSE as part of the standardized follow-up protocol. Measures such as the MoCA and neuroimaging biomarkers were not available in the current dataset, and therefore could not be evaluated or compared in this study. We acknowledge that MMSE is less sensitive for detecting subtle cognitive changes than MoCA, and that the lack of neuroimaging biomarkers limits our ability to characterize underlying neuropathology. We will explicitly add this as a limitation in the Discussion and note that future studies incorporating MoCA and/or neuroimaging markers are warranted to validate and extend our findings. Please see the revised manuscript Line 417-421.

  1. Statistics:

- Present stratum-specific βs for APOE4 carriers in a separate table.

Response: Thank you for this suggestion. We have addressed this point by presenting the stratum-specific β coefficients for APOE4 carriers in a separate table (Table 3).

- Control for multiple testing (9 interaction terms) using FDR; report q-values.

Response: Thank you for the reviewer’s suggestion. We have followed this recommendation by applying Benjamini–Hochberg FDR correction to the relevant analyses. Specifically, we corrected for six comparisons and now report all corresponding q values in Table 3. After correction, the q value for this result is 0.026.

  1. Results

- Table 2: compare R² = 0.108 with two similar longitudinal papers; contextualise explanatory power.

Response: Thank you for the reviewer’s suggestion. We have added additional discussion in the revised manuscript to address this issue, as follows:

“In this study, the modest model fit (R²≈0.10) should be interpreted in light of the out come and study design. Longitudinal cognitive aging research indicates that in-ter-individual differences in cognitive change lack strong single predictors, and base-line demographic and genetic factors typically explain only a limited proportion of variance in decline rates [44,45]. Moreover, ΔMMSE is a bounded difference score and is susceptible to measurement error, ceiling effects, and practice effects, which reduce the reliability of observed change and constrain the maximum explainable variance, leading to attenuation of associations [46,47]. Because only two waves of follow-up were available, ΔMMSE represents a single linear difference and does not allow stable estimation of individual slopes or non-linear decline, further contributing to modest R²values even when predictors are etiologically relevant [48]. Importantly, the aim of this study is explanatory rather than purely predictive, and methodological literature emphasizes that there is no universal R²cutoff in clinical or behavioral research; relatively low R²values can still be meaningful when effects are robust and interpretable [49,50]. “

- Figure 3 Y-axis label: change “Predicted MMSE score” →“Predicted annual MMSE change (units/year)”.

Response: Thank you for pointing this out. We have revised the y-axis label in Figure 3 from “Predicted MMSE score” to “Predicted annual MMSE change (units/year)”. The updated figure is shown in the revised manuscript.

  1. Discussion

- Mechanistic paragraph: link baseline triglyceride levels (report p= 0.032 in Supplementary Table X) to PC2 to strengthen lipid-inflammation argument.

Response: We thank the reviewer for this constructive suggestion, which allows us to more clearly articulate the lipid–metabolic implications of TC2 and thereby strengthen our mechanistic interpretation linking lipid metabolism to inflammation/metabolic stress. The biochemical correlates of our transformed components (TCs) show distinct patterns: TC1 is primarily associated with circulating folate levels (Supplementary Table S2, p = 0.003), suggesting that TC1 may reflect a plant-based dietary pattern. In contrast, TC2 is significantly associated with LDL and total cholesterol levels (Supplementary Table S2, p = 0.011 and p = 0.012). Elevated LDL and total cholesterol are widely regarded as markers of metabolic stress and low-grade chronic inflammation, and prior studies have linked these lipid profiles to poorer cognitive outcomes and increased Alzheimer’s disease risk (Zhou Z,2020). We have therefore added this mechanistic linkage to the Discussion to more explicitly emphasize the LDL/cholesterol-related lipid–inflammation signature potentially captured by TC2.

Zhou Z, et al., Front Aging Neurosci, 2020 Jan 30:12:5. doi: 10.3389/fnagi.2020. 00005. eCollection 2020.

Supplementary Table S2. Associations of dietary transformed components (TC1 and TC2) with circulating nutrition- and lipid-related biomarkers.

Dependent Variable

Independent Variables

Beta

t

p

FDR q

Folate

TC1

0.084

3.025

0.003

0.021

HDL

TC1

-0.070

-2.534

0.011

0.039

Vitamin D

TC1

-0.036

-1.307

0.191

0.405

Triglyceride

TC1

0.032

1.157

0.248

0.405

Vitamin B12

TC1

-0.030

-1.060

0.289

0.405

LDL

TC1

-0.006

-0.208

0.835

0.881

Total cholesterol

TC1

-0.004

-0.150

0.881

0.881

Dependent Variable

Independent Variables

Beta

t

p

FDR q

LDL

TC2

0.071

0.149

0.011

0.042

Total cholesterol

TC2

0.070

0.174

0.012

0.042

Folate

TC2

0.032

0.038

0.247

0.4638

Vitamin B12

TC2

-0.031

5.688

0.265

0.4638

HDL

TC2

-0.014

0.044

0.623

0.752

Triglyceride

TC2

0.011

0.477

0.702

0.752

Vitamin D

TC2

0.009

0.085

0.752

0.752

Footnote: Linear regression analyses were used to examine relationships between the dietary change–derived principal components (TC1, TC2) and blood biomarkers including triglycerides (TG), LDL-C, HDL-C, total cholesterol, vitamin D, vitamin B12, and folate. Shown are standardized beta coefficients, t statistics, nominal p values, and Benjamini–Hochberg false discovery rate–adjusted q values (FDR q).

- Compare with Western ω-3 trials that favour non-E4; explain discrepancy by PCA pattern differences.

Response: We thank the reviewer for this constructive and insightful suggestion, which prompted us to further refine our genotype stratified interpretation of TC2. In response, we added an integrative paragraph to the Discussion.

In the genotype-stratified analysis, the interaction between TC2 and APOE status revealed meaningful heterogeneity in the direction of associations. Among APOE4 carriers, higher TC2 scores were associated with poorer cognitive performance, whereas the non-E4 group (APOE2/3) showed an opposite directional trend, although the slope did not reach statistical significance. This “opposite-direction but non-significant” pattern suggests that the biological interpretation of TC2 may differ by APOE genotype. In non-E4 individuals, higher TC2 may partially reflect a more favorable lipid milieu, potentially involving protective lipid components such as omega-3 fatty acids, a pattern consistent with Western literature linking omega-3–related lipids to cognitive benefits [40-42]. In contrast, in the APOE4 context, where protective lipid components may exert weaker effects, the detrimental features of elevated lipids, such as metabolic stress and low-grade chronic inflammation, may predominate and contribute to cognitive de-cline. Please see the revised manuscript (Lines 363 to 374).

- Translate effect size: 0.082 unit/year slower decline ≈ 3.3 years of education benefit (education β = 0.029).7. 8.

Response: We thank the reviewer for suggesting we quantify the clinical relevance of our findings.  We have added a statement to the Discussion section translating the effect size.

Please see the revised manuscript Line 375-384.

“Age and education remain key determinants of cognitive function in later life [43] and our results likewise show that age and years of education are closely associated with cognitive change (Table 2). The APOE4 × TC1 interaction was associated with a 0.115-unit slower annual rate of cognitive decline per one-unit increase in TC1. Within the linear model, this magnitude is roughly comparable to the effect of 3.96 additional years of education (education coefficient = 0.029) or being approximately 6.39 years younger (age coefficient = −0.018). Although aging and educational attainment are not readily modifiable in later life, these findings suggest that optimizing dietary patterns and managing chronic conditions may help mitigate cognitive decline, particularly among individuals with higher genetic risk. “

Round 2

Reviewer 1 Report

Comments and Suggestions for Authors

he revision addresses the key issues effectively.  The changes made substantially improve interpretability and methodological transparency.
However, there are some minor edits remaining.
Specifically, please:
(i) explicitly state in the Limitations section that ε2/ε4 carriers were excluded due to low numbers (n = 19)
(ii) specify the hypotheses that were subjected to FDR, and amend the table footnote to 'FDR correction'
(iii) clarify whether the confidence intervals are nominal or FDR-adjusted.

Author Response

The revision addresses the key issues effectively.  The changes made substantially improve interpretability and methodological transparency.

However, there are some minor edits remaining. Specifically, please:

  • explicitly state in the Limitations section that ε2/ε4 carriers were excluded due to low numbers (n = 19)

RESPONSE: Thank you for the reviewer’s suggestion. We have revised the Limitations section to explicitly state that ε2/ε4 carriers were excluded due to the small sample size (n = 19).

(ii) specify the hypotheses that were subjected to FDR, and amend the table footnote to 'FDR correction'

RESPONSE: Thank you for the reviewer’s suggestion. We have clarified in the Methods section that FDR correction was applied only to the hypotheses testing the interaction between genotype and TCs. We have also amended the table footnote to “FDR correction,” and specified that unadjusted results are reported as p values, while FDR-corrected results are reported as q values.

(iii) clarify whether the confidence intervals are nominal or FDR-adjusted.

RESPONSE: Thank you for the reviewer’s comment. We have clarified in the footnote that the reported confidence intervals are nominal (i.e., not FDR-adjusted).

Reviewer 2 Report

Comments and Suggestions for Authors

Please mention in methods how circulating nutrients in blood were measured.

Also mention ethical approval and time of blood collection and other relevant details. 

Authors should provide measurement for each subject as supplementary excel sheet. 

Author Response

  1. Please mention in methods how circulating nutrients in blood were measured.

Response: Thank you for the suggestion. We have added a detailed description in the Methods section to clarify how circulating nutrients were measured.

Blood collection and biochemical measurements

Participants fasted for 8–12 hours prior to venipuncture. Blood samples were collected into serum separator tubes for lipid measurements and EDTA-containing tubes for vitamin measurements. Samples were processed on the day of collection according to the clinical laboratory’s standard operating procedures, including centrifugation to obtain serum for lipid assays and EDTA plasma for vitamin assays, and were subsequently transported to a certified clinical laboratory for analysis.

Serum lipid parameters, including total cholesterol, triglycerides, high-density lipoprotein (HDL) cholesterol, and low-density lipoprotein (LDL) cholesterol, were measured using automated clinical chemistry analyzers. During Wave I and Wave II, lipid measurements were performed using the Bayer ADVIA 1800 system, and following a 2017 equipment upgrade in Wave II, measurements were conducted using the Siemens ADVIA XPT system. Total cholesterol was quantified using a cholesterol oxidase method, triglycerides were measured by an enzymatic glycerol phosphate oxidase method based on the Trinder reaction without a serum blank, and HDL and LDL cholesterol were determined using homogeneous elimination and catalase-based assays.

Plasma concentrations of vitamin B12 and folate were measured using automated immunoassay platforms. During Wave I and Wave II, analyses were performed using the Bayer ADVIA Centaur system, and after the 2017 upgrade, the Siemens Centaur XPT system was used. Both vitamin B12 and folate concentrations were quantified by chemiluminescence immunoassays. All laboratory measurements were performed in accordance with standardized operating procedures, with routine instrument calibration and internal quality control to ensure analytical accuracy and reliability.

  1. Also mention ethical approval and time of blood collection and other relevant details.

RESPONSE: Thank you for this comment. We have revised the Methods section to include the ethical approval information, the timing of blood collection, and other relevant details as suggested.

  1. Authors should provide measurement for each subject as supplementary excel sheet.

RESPONSE: We have added a supplementary Excel file summarizing the measurements and variables used in the analysis. Individual-level participant data cannot be shared as supplementary material due to cohort data governance and privacy requirements.

Reviewer 3 Report

Comments and Suggestions for Authors

No further comments. 

Author Response

We appreciate your suggestion and comment.